# IBiT: Utilizing Inductive Biases to Create a More Data Efficient Attention Mechanism

## Abstract

In recent years, Transformer-based architectures have become the dominant method for Computer Vision applications. While Transformers are explainable and scale well with dataset size (Dosovitskiy et al., 2020), they lack the inductive biases of Convolutional Neural Networks (LeCun et al., 1989). While these biases may be learned on large datasets, we show that introducing these inductive biases through learned masks allow Vision Transformers to learn on much smaller datasets without Knowledge Distillation. These Transformers, which we call Inductively Biased Image Transformers (IBiT), are significantly more accurate on small datasets, while retaining the explainability Transformers.

## 1 Introduction

Vision Transformers (Dosovitskiy et al., 2020) have become established as the state-of-the-art Computer Vision backbone. Such self-attention (Vaswani et al., 2017) based networks have also shown excellent pre-training performance on datasets like ImageNet (Deng et al., 2009). Unfortunately, Vision Transformers require large amounts of data (14M-300M images) during pre-training to outperform Convolutional Neural Networks (Dosovitskiy et al., 2020). In more specialized domains where transfer learning on large image recognition datasets is not possible, and dataset size is often quite small (Zhai et al., 2020), Convolutional Neural Networks (CNNs), more specifically ResNets (He et al., 2015), are still the dominant model architecture.

This is because CNNs possess inductive biases, namely translational equivariance and locality (Dosovitskiy et al., 2020), which allow such models to achieve better results with smaller amounts of data.

Using this intuition, we propose Inductively Biased Image Transformers, or IBiTs, a model that can approximate the inductive biases of a CNN, allowing it to learn better on small datasets. This is possible through the application of learnable masks to the attention layers during self-attention (Vaswani et al., 2017), allowing IBiTs to mimic a CNN's biases with a Transformer architecture. Using a technique called Rank Approximation, we significantly reduce the parameters required to implement these learnable masks by using low rank approximations.

When all of these methods are implemented, the final model is able to outperform comparable Transformer-based methods (Touvron et al., 2021) by two percentage points when trained solely on ImageNet, achieving state-of-the-art image transformers performance while retaining explainability and scalability.

## 2 Related Work

Vision transformers were first introduced by Dosovitskiy et al. (2020) and pretrained Vision Transformers have seen use in a wide variety of applications.

DeiTs were the first models to explore generalizing Transformers for smaller datasets using knowledge distillation. Knowledge distillation uses strong teacher models to introduce inductive biases into Vision Transformers. Although we also introduce inductive biases, our method, which uses learnable masks, does not use strong teacher models which often need large amounts of data and computational power to train.

Another method to induce inductive biases in Transformers models involves the use of learned relative positional encodings to make self-attention layers have the capacity to model CNN layers (Cordonnier et al., 2020). However, this method does not perform as well as Vision Transformers. It also requires the number of heads in the self-attention layer to be equivalent to the square of the filter size being approximated (eg. A transformer would need 9 heads to approximate a convolutional layer with a filter size of 3x3).

To improve the performance of relative positional encodings, further research involved using a weighted sum between relative positional encodings and normal self-attention (d'Ascoli et al., 2022). This method outperformed DeiTs by 0.9 percent. However, ConViTs, as the models are called, slightly modify the architecture of DeiTs, using four heads instead of the three normally used by DeiTs.

Our method, IBiT, does not use relative positional encodings, instead using learnable masks. By using this new approach, IBiTs achieve improved performance with fewer parameters and faster performance. IBiTs do not require any underlying assumptions surrounding model architecture like ConViTs do, meaning it is easy to modify existing models to incorporate our methods, by just converting a normal self-attention layer into a learned mask self attention layer with the exact same structure. Our method also performs significantly better than both these approaches of introducing inductive biases into Transformers, while retaining the explainability of attention-based models.

## 3  MODEL ARCHITECTURE

### 3.1  APPROXIMATING CONVOLUTION WITH SELF-ATTENTION

To approximate convolution with self-attention, we begin with convolution with a single filter and a single channel.

$$Y_{i,j} = \sum_{k=0}^{f} \sum_{l=0}^{f} X_{(i+k),(j+l)} * W_{k,l}$$

In this equation, $f$ represents the filter size, $W$ represent the filter weights, $X$ is a 2-D input image, and $Y$ is the result of the convolution operation.

The equation for a single head of self-attention is as follows:

$$Y_i = \sum_{m=0}^{size} X_m * W_{i,m}$$

where $X$ is a 1-D representation created by flattening the image, $size$ is the length of the 1-D representation of the image, and $W$ is the attention map generated by self attention.

By re-indexing $size$ in terms of $height$ and $width$ and re-indexing $i$ in terms of $i$ and $j$ in the 2-D image, we can create the following equation, which is equivalent to self-attention for a single head.

$$Y_{i*width+j} = \sum_{m=0}^{height} \sum_{n=0}^{width} X_{m*width+n} * W_{i*width+j,m*width+n}$$

Setting the row $W_{i*width+j}$ to equal $W_{k,l}$ where $k = m \in \{0, 1, ..., f\}$ and $l = n \in \{0, 1, ..., f\}$, and setting all other entries in the matrix $W$ to zero, the equation simplifies to the equation below.

$$Y_{i*width+j} = \sum_{k=0}^{f} \sum_{l=0}^{f} X_{(i+k)*width+(j+l)} * W_{k,l}$$

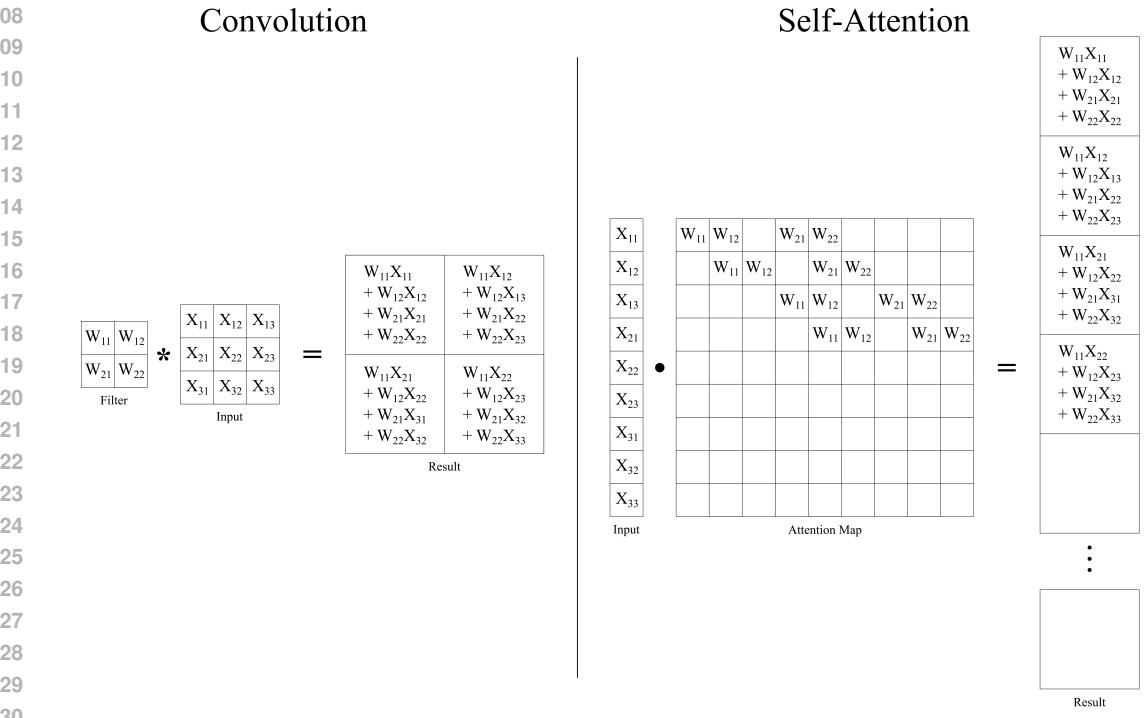

Figure 1: Visual Representation of Convolution through Matrix Multiplication

This equation is equivalent to the equation for convolution, where the output is flattened into a 1-D representation. For a visual representation of the process, see Figure 1.

### 3.2 LOW RANK APPROXIMATION

To introduce inductive biases into self-attention the weights of the attention map at row $i * width + j$ should be equal to $W_{k,l}$ where $k = m \in \{0, 1, ..., f\}$ and $l = n \in \{0, 1, ..., f\}$.

This means that the attention map has a rank of $height * width$. In order for the queries and keys to reproduce an attention map with inductive biases, both the queries or the keys need to have a rank of $height * width$. Since the maximum rank of a matrix is the minimum of the number of rows or columns, and the sequence length of the unrolled image is $height * width$, the embedding size of the matrix must be equal to or larger than the sequence length for inductive biases to arise in the self-attention layer.

In practice, this is rarely the case. Increasing the embedding size increases the computational cost, and most common architectures have significantly larger sequence lengths than embedding sizes. Dedicating such a large proportion of the network to represent inductive biases would also reduce the performance of the model.

To mitigate these problems, we introduce rank approximation. Due to the sparsity of the attention map, by rolling each row back by its row number, we are able to reduce the rank of the attention map to the square of the filter size (i.e., a $3 \times 3$ filter could be represented by a matrix with rank 9).

Since inductive biases do not depend on the content of the image, we use two separate learnable matrices called sub-masks. These matrices have a shape of $(height * width) \times mask\_fidelity$, where mask fidelity is the square of the filter size. Through a dot product, these sub-masks, A and

B respectively, become the inductive bias matrix.

$$W^{inductive} = A \bullet B^T$$

We then undo the rank-reducing rolling operation and multiply this inductive bias matrix and the attention map element-wise, essentially treating the unrolled inductive bias matrix as a mask on the attention map.

This ensures that our inductive biases are parameter-efficient and persistent regardless of image content.

### 3.3  MASK TRAINING

We initially set the inductive biases to an inductively biased attention map. Instead of training our masks to initially represent a Convolutional kernel, we use the same rank approximation methodology to approximate a Gaussian kernel.

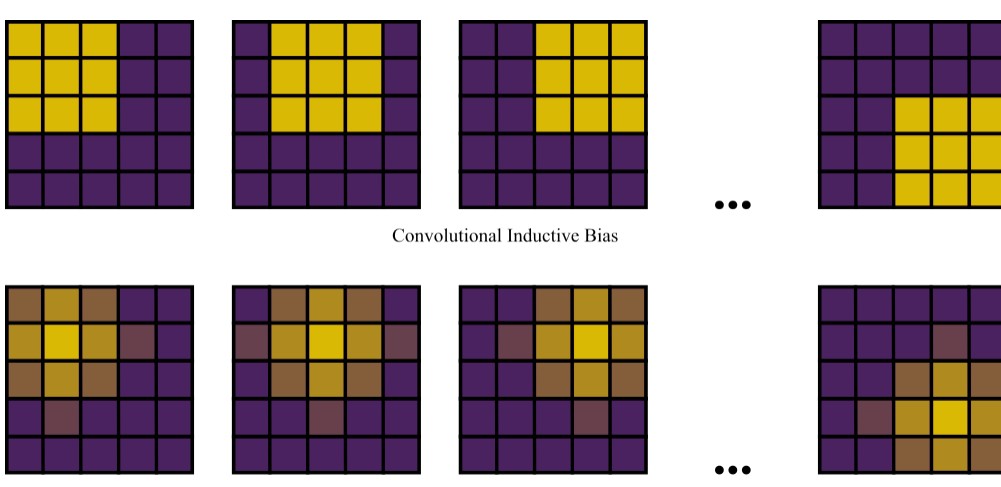

Figure 2: Convolutional Inductive biases versus Gaussian Kernel Inductive Biases

To do this, the initial weight matrices are trained to approximate the unrolled attention map using the algorithm below.

---

**Algorithm 1** Mask Weights Training Procedure

---

1: **procedure** TRAINMASKWEIGHTS(x, num_heads, d_model)
2:    $A \leftarrow$ RANDOMINITIALIZATION($mask\_fidelity, height * width$)     ▷ Initialize first submask, A, with shape ($mask\_fidelity, height * width$)
3:    $B \leftarrow$ RANDOMINITIALIZATION($mask\_fidelity, height * width$)     ▷ Initialize second submask, B, with shape ($mask\_fidelity, height * width$)
4:    $\alpha \leftarrow 0.1$     ▷ Initialize learning rate to high value of 0.1
5:    $filter\_matrix \leftarrow$ RADIALATTENTIONMATRIX($filter$)     ▷ Initialize Target Attention Map(Fig. 1)
6:    **for** $i = 1, \ldots, epochs$ **do**
7:       $output \leftarrow A^T \bullet B$
8:       $loss \leftarrow$ MSE($filter\_matrix, output$)
9:       $A \leftarrow A - \alpha \cdot \frac{\partial loss}{\partial A}$     ▷ Take Gradient Descent step for $A$
10:      $B \leftarrow B - \alpha \cdot \frac{\partial loss}{\partial B}$     ▷ Take Gradient Descent step for $B$
11:    **end for**
12:    **return** $(A, B)$
13: **end procedure**

---

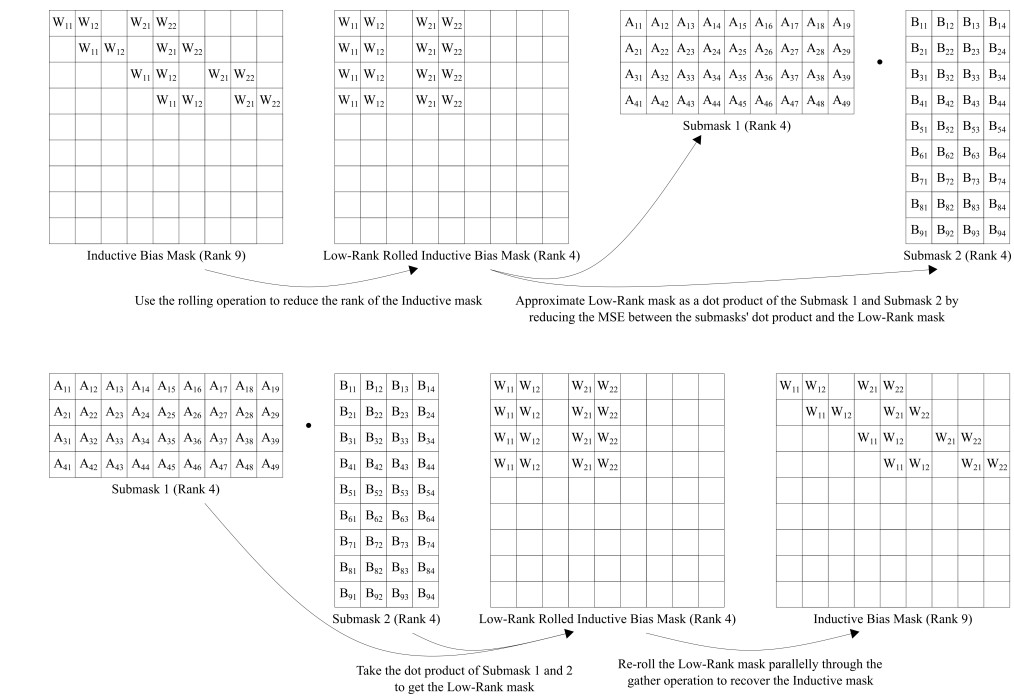

Figure 3: Visual Representation of LMSA Layer

The algorithm for the full Learned Mask Self Attention layer is described in the page below, while Figure 3 presents a visual representation of the Learned Mask Self Attention process.

---

**Algorithm 2** LMSA Layer

1: **procedure** LMSA(x, batch_size, seq_len, num_heads, d_model, w_mask1, w_mask2)
2:     $keys \leftarrow \text{LINEAR}(W_{keys}, x)$    ▷ Linearly transform input to get keys, which has a shape of (batch_size, seq_len, d_model), where seq_len is and image's height times its width
3:     $queries \leftarrow \text{LINEAR}(W_{queries}, x)$
4:     $values \leftarrow \text{LINEAR}(W_{values}, x)$
5:     $keys \leftarrow keys.\text{RESHAPE}(batch\_size, seq\_len, num\_heads, \frac{d\_model}{num\_heads}).T$
6:     $queries \leftarrow queries.\text{RESHAPE}(batch\_size, seq\_len, num\_heads, \frac{d\_model}{num\_heads}).T$
7:     $values \leftarrow values.\text{RESHAPE}(batch\_size, seq\_len, num\_heads, \frac{d\_model}{num\_heads}).T$    ▷ Split keys, queries, values along channels to get shape of (batch_size, num_heads, seq_len, $\frac{d\_model}{num\_heads}$)
8:     $W^{inductive} \leftarrow A^T \bullet B$
9:     $W^{inductive} \leftarrow \text{ROLL}(W^{inductive})$    ▷ Rolls w_mask by row number. More details on procedure in Appendix A
10:     $W^{attention} \leftarrow keys \bullet queries.T$
11:     $W^{new\_attention} \leftarrow W^{attention} \times W^{inductive}$ ▷ Has shape (batch_size, num_heads, seq_len, seq_len)
12:     $W^{new\_attention} \leftarrow \frac{W^{new\_attention}}{\text{NORM}(W^{new\_attention})}$    ▷ Normalizes $W^{new\_attention}$
13:     $output \leftarrow W^{new\_attention} \bullet values$    ▷ Has shape (batch_size, num_heads, seq_len, $\frac{d\_model}{num\_heads}$)
14:     $output \leftarrow output.T.\text{RESHAPE}(batch\_size, seq\_len, d\_model)$    ▷ Reshape back to same dimensions as input
15:     **return** $output$
16: **end procedure**

---

## 4 EXPERIMENTS

### 4.1 EXPERIMENTAL SETUP

Our model architecture is the same as the DeiT model architecture, with the same number of heads per layer, and the same number of layers. We replace all Multi-Head Self Attention layers with Learned Masked Self Attention. We also remove the CutMix data augmentation technique (Yun et al., 2019), because it hinders the learning of inductive biases. Finally, we remove stochastic depth on the feed-forward layer, since it improves ViT performance. We also compare our results to ConViTs, which have one more head than both DeiTs, and our model. We detail the hyper-parameters and other model information in the table below. We train our models on ImageNet-1k for 300 epochs, and use the reported results on ImageNet-1k for other models.

Table 1: Training Configurations

| Model Type | # of Layers | # of Heads | $D_{model}$ | Learning Rate | # of Params |
|---|---|---|---|---|---|
| IBiT (Ours) | 12 | 3 | 192 | 0.001 | 6M |
| DeiT | 12 | 3 | 192 | 0.001 | 6M |
| ConViT | 12 | 4 | 192 | 0.001 | 6M |

For our scaling experiments, we compare DeiTs and ConViTs on varying percentages of the ImageNet dataset to show dataset sample efficiency.

Table 2: Scaling Configurations

| Model Type | # of Layers | # of Heads | $D_{model}$ | Learning Rate | # of Params |
|---|---|---|---|---|---|
| IBiT (Ours) | 12 | 3 | 192 | 0.001 | 6M |
| DeiT | 12 | 3 | 192 | 0.001 | 6M |

To show explainability, we use Rollout Attention to highlight which pixels the model is using to classify the input (Abnar & Zuidema, 2020).

### 4.2 RESULTS

Here we present the results of our experiments. Our model, the IBiT, outperforms both DeiTs and ConViTs significantly. The scaling curve indicates that our model is even more scalable than a DeiT, improving at a faster rate as dataset size increases. This is in contrast to previous approaches of inducing inductive biases, which mainly show large increases at very small dataset sizes.

Table 3: Training Results

| Model Type | Accuracy | # of Params |
|---|---|---|
| IBiT (Ours) | **74.2** | 6M |
| DeiT | 72.2 | 6M |
| DeiT (Our Reproduction) | 71.5 | 6M |
| ConViT | 73.1 | 6M |

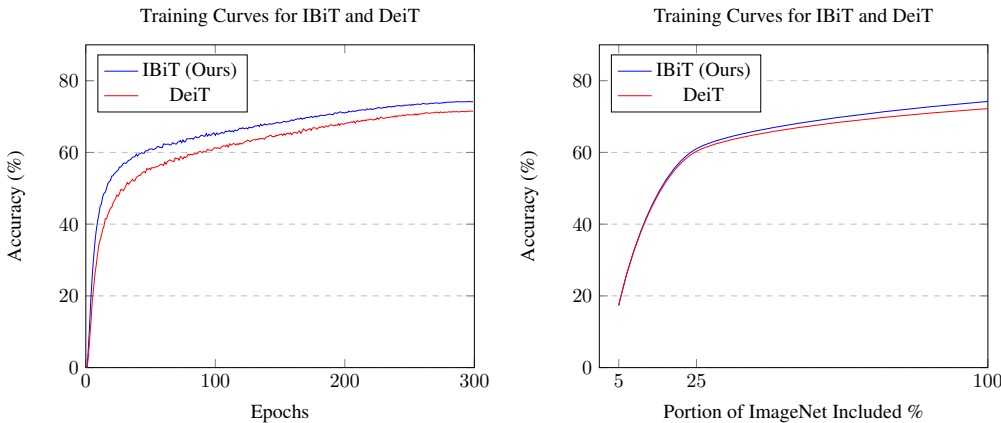

Figure 4: Training and Scaling Curves for IBiT and DeiT

## 4.3 EXPLAINABILITY

Here are some representative attention maps between the CLS token and the image, using Rollout Attention (Abnar & Zuidema, 2020). More attention maps can be found in Appendix B.

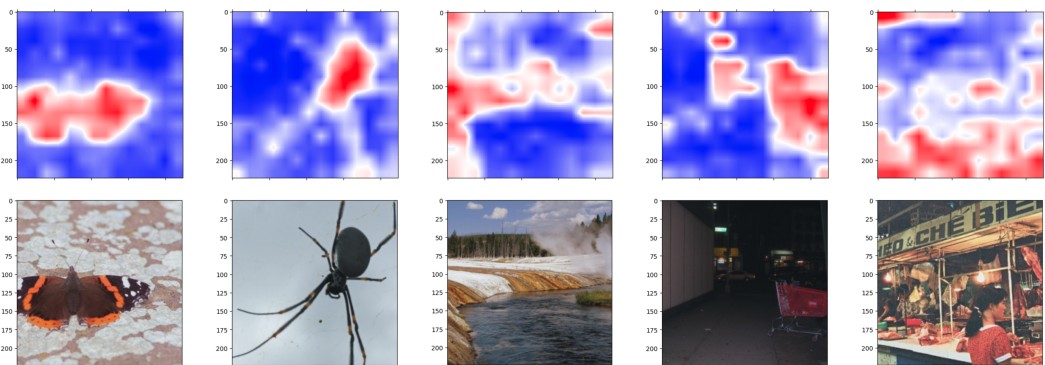

Figure 5: Representative Attention Maps using Rollout Attention

The model typically attends to the subject of the image. Through attention maps, we show that the explainability of Transformers is preserved, despite the use of Learnable Masks.

## 5 ABLATION RESULTS

Here we present the results for our different ablation configurations. for our ablation, we ablate on

Table 4: Ablation Configurations

| Configuration | Learnable Masks | CutMix | Accuracy |
|---|---|---|---|
| A (IBiT) | ✓ | ✗ | 74.2 |
| B | ✓ | ✓ | 72.2 |
| C (DeiT) | ✗ | ✓ | 71.5 |

the Learnable Mask Self Attention Layer, and on CutMix. We find that the use of CutMix degrades model performance significantly. By mixing different portions of the image, CutMix hinders the formation of inductive biases in our model, leading to reduced performance.

# 6 DISCUSSION

## 6.1 LEARNABLE MASKS

Here we show the learnable masks as they evolve throughout the training process. You can see how later layers have less of the initial inductive biases, while earlier layers learn multiple strong inductive biases quickly.

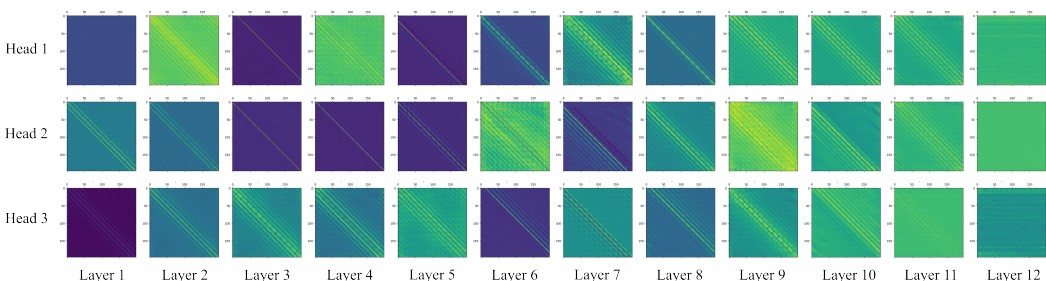

Figure 6: Learnable Mask Visualization for Trained IBiT Network

This motivates the idea that learned masks may not even be needed in later layers, though we leave further architectural modifications to future work.

## 6.2 GRADIENT BIASING

One reason we believe models using learnable masks perform better is the property of gradient biasing.

Since we set the mask to mimic the inductive biases of a Gaussian kernel, and the mask is applied through element-wise multiplication, the gradient update on the attention mask can be calculated as $W^{mask} \frac{\partial J}{\partial W^{attention}}$

This means that at each weight of the mask, the gradient, $\frac{\partial J}{\partial W^{attention}}$ is scaled by the mask weight. When this mask represents a gaussian kernel, this means that values close to each are scaled to have a larger gradient, and hence the model learns to use these values better. These inductive biases are visible in the learnable mask even during late stages of training (Figure 5 and Appendix C) highlighting the effect of gradient biasing.

# 7 CONCLUSION

Clearly, using learnable masks preserves the explainability of regular Transformers, while being significantly more accurate on ImageNet. These models are significantly more sample efficient compared to other other Transformer based models. We hope the interesting new properties these models possess and further work on parameter efficiency and more efficient learnable masks may lead to significant advancements in the field of explainable, data-efficient computer vision.

# 8 REPRODUCIBILITY

To ensure that our results are reproducible, we have published our trained models and code on GitHub. We have also put all model hyperparameter configurations for all the model training runs in the appendices.

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

# APPENDIX

## A  MORE EXPLAINABILITY VISUALIZATIONS

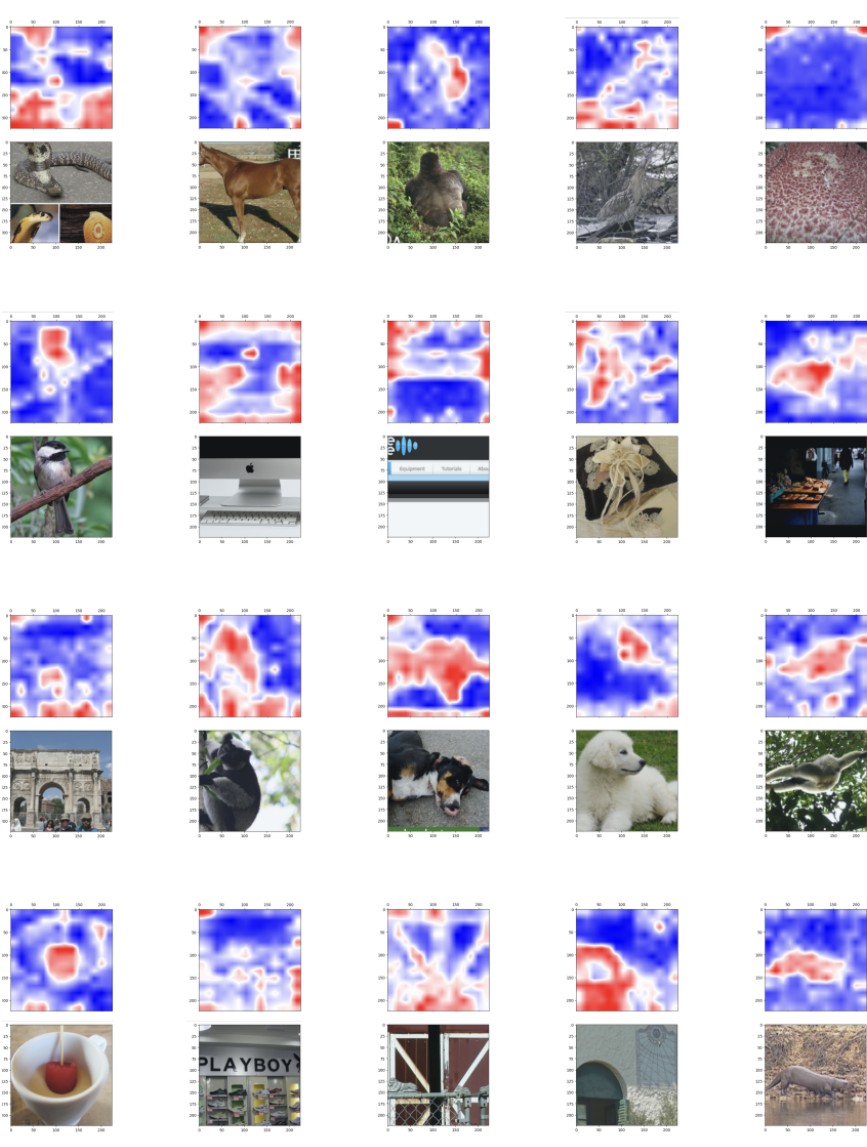

Figure 7: More Representative Attention Maps using Rollout Attention

## B    MASK VISUALIZATIONS THROUGH LEARNING

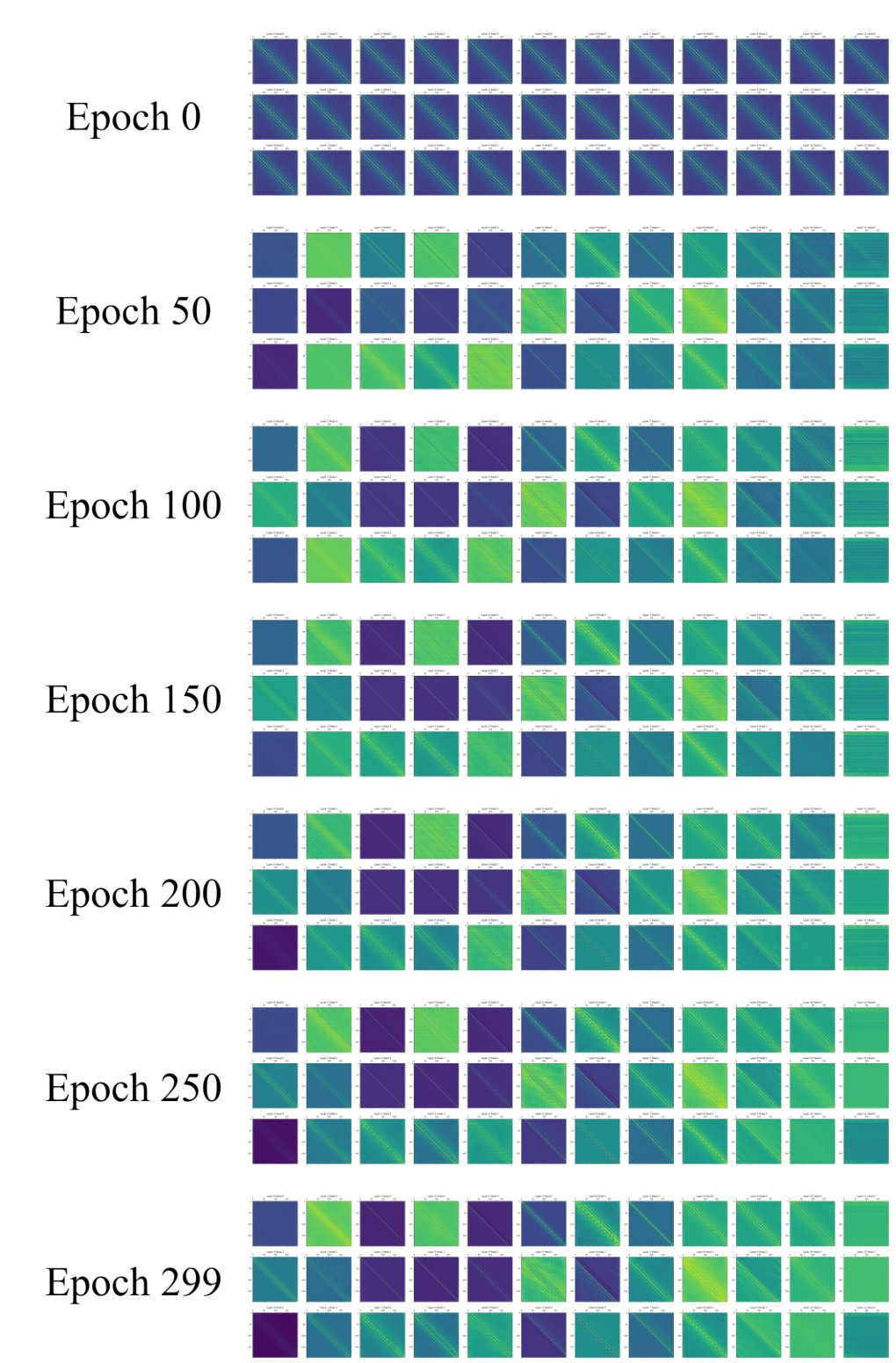

Figure 8: Learnable Mask Visualization during Training of IBiT Network

# C HYPERPARAMETER CONFIGURATIONS FOR TRAINED MODELS

Table 5: Training Configurations

| Model Type | IBiT (Ours) | DeiT (Our Reproduction) |
|---|---|---|
| **# of Layers** | 12 | 12 |
| **# of Heads** | 4 | 4 |
| **$D_{model}$** | 192 | 192 |
| **Learning Rate** | 0.001 | 0.001 |
| **Weight Decay** | 0.005 | 0.005 |
| **Label Smoothing Rate** | 0.1 | 0.1 |
| **Dropout Rate** | 0.0 | 0.0 |
| **Drop Path Rate** | 0.1 | 0.1 |
| **Learning Rate Scheduler** | Cosine-Warmup | Cosine-Warmup |
| **# of Params** | 6M | 6M |
| **CutMix $\alpha$** | N/A | 1.0 |
| **MixUp $\alpha$** | N/A | 0.8 |

