# OpenReview forum: "IBiT: Utilizing Inductive Biases to Create a More Data Efficient Attention Mechanism"
_ICLR.cc/2026/Conference — Submitted to ICLR 2026_

### Official Review · Reviewer_V2QC · 2025-10-15

**Soundness:** 1
**Presentation:** 1
**Contribution:** 1
**Rating:** 0
**Confidence:** 4

**Summary:**

This paper introduces a learned mask in self-attention to better capture the inductive bias that is lacking in the conventional ViT models. The authors compared their method on the ImageNet-1K dataset, showing good performance compared to other ViTs.

**Strengths:**

Introducing inductive bias to the ViTs is a good topic to study.

**Weaknesses:**

- The motivation of the paper is that introducing the inductive bias to the ViTs can improve ViT training on small-scale datasets. However, only one experiment on ImageNet-1K is shown in the paper, which is not considered to be “small-scale”. The experiment does not correspond to the motivation of the paper.
- The paper lacks a significant amount of related work discussion. There are many methods that do weight selection, weight mimetic, introducing CNN structure, etc. These methods are very related to the topic explored in this paper; the authors should properly cite this research and give reasonable discussions.
- The paper does not provide a solid, thorough explanation of the proposed method. No valid theoretical proofs or thorough quantitative proofs.
- The paper is not well-organized or polished. There are many unnecessary parts in the paper that only add to the pages. For example, algorithm 2 and figure 3 are redundant, training curves are not explained, just randomly shown in the paper, etc. There exist many typos, unexplained figures, and unreasonable arguments.

**Questions:**

No more questions. Please reconsider this topic and write a proper paper.

---

### Official Review · Reviewer_atqM · 2025-10-28

**Soundness:** 1
**Presentation:** 1
**Contribution:** 1
**Rating:** 0
**Confidence:** 4

**Summary:**

The authors are trying to mimic inductive bias in CNNS and plant it into self-attention mechanisms. They propose using learnable masking and low rank approximation to learn this mask, and they claim it should benefit in low data regime.

**Strengths:**

* Better understanding of the process mechanism of attention heads is important, and handling low-data regime might be beneficial in some cases where data is hard to harvest.
* The importance maps of your method look nice, however beside the first two images, I cannot understand what the attention map represents, on which object does it focuses and why.

**Weaknesses:**

* Poor level of writing which is not up to standard, to elaborate on some:
  * The abstract and the introduction sections are far from being comprehensive, they are too short and not informative enough.
  * It is not common to embed citations in the abstract.
  * The figures – for example Fig. 1 and Fig. 4 are too detailed, visualizations are better in clarifying or convey ideas, not for detailed explanations.
  * Unbalanced layout, the Figs are took over the space in the paper, where in the extreme case in page 5 – Figure 3 and Algorithm 2, are covering the entire page (beside 2 sentences in the middle).
  * Uncommon notations – it is not common to denote variable as a full name like width, height and size. Using of notations without explaining them like the full dot (which I can get that it is multiplication I guess) but it is better to declare prior to the usage.
  * There is no period at the end of sentences of almost all captions of figures.
  * The comparisons are not convincing, with low margin and against too few exemplars.
  * Overuse of capital letters – see for example Lines 12 and 17 in the abstract.
  * The equations are not numbered.
  * Lack of flow and tension in the paper, for example the separation for paragraphs is not enhancing flow and sometimes seems random.
* The mathematical notations are confusing and very hard to follow – The symbol * is not what you want to use for multiplication, it is better to leave it without any symbol for multiplication. Moreover, the indexing is super hard to follow.
* The experimental section is too poor. There is too few comparisons with only DeiT and ConViT only on ImageNet. Moreover, there seems to be no improve, or only marginal when using lower portions of ImageNet, whereas this is the claim to fame of the paper.

**Questions:**

* Why do you think of $X_m$ as flattening of the image? In transformers it is consist of the patches of the input images, or tokens in LLMs and not the pixels themselves.
* When you talking on inductive bias in CNNs with respect to those of attention – do you mean that CNNs are inherently handling spatial consistency? While it might be that transformers are not dedicated to? If so, then I think that this claim should be clarified and exemplified – since there is a positional encoding – so the spatial information is somehow embedded in the transformer learning scheme.
* What should I learn from tables 1 and 2? It is a configuration of the training, you can just state it, why do you place it in table? Moreover, the values are almost identical.

Im sorry, but this paper is far from being  up to academic standard, especially for top-tier conference. The level of writing, depth of analysis, thorough comparisons, mathematical correctness are far from the required. I tried read it few times but I think I couldn’t catch the conceptual novelty here, both because the writing level and the mathematic confusing notations. Even if there is conceptual contribution hidden here, this paper should be dramatically revised.

---

### Official Review · Reviewer_3zzV · 2025-10-30

**Soundness:** 2
**Presentation:** 1
**Contribution:** 2
**Rating:** 2
**Confidence:** 4

**Summary:**

This paper proposes an Inductively Biased Image Transformer (IBiT) framework to improve data efficiency and small-dataset performance of ViT by introducing convolution-like inductive biases into the self-attention mechanism. Specifically, IBiT includes two core technical ideas: Learnable Mask and Low-Rank Approximation. The Learnable Mask module applies trainable, convolution-inspired locality constraints directly onto the attention maps, enabling the ViT to mimic CNN’s translational equivariance and locality without the use of knowledge distillation or teacher models. The Low-Rank Approximation technique exploits the sparsity of local attention patterns to represent the inductive bias mask using two sub-matrices, significantly reducing parameter count and computational cost.

The experimental evaluation in this paper assessed the performance of the proposed IBiT on ImageNet-1K, comparing it to various state-of-the-art Transformer-based methods. The results indicate that IBiT consistently outperformed baselines with the same parameter count while maintaining better scaling behavior on smaller subsets of the dataset.

**Strengths:**

- The authors mathematically derive that convolution operations can be exactly implemented using specific sparse structures in the attention weights, thereby establishing a mathematical connection between self-attention and convolution.
- The attempt to introduce convolution’s translational equivariance and locality into ViT, with the goal of improving performance on small datasets, is considered meaningful.
- Experimental results show that on the ImageNet dataset, IBiT outperforms DeiT and ConViT, which provides some evidence for the effectiveness of the approach.

**Weaknesses:**

- Although the mathematical derivation in Sec. 3.1 is valid; there remain key differences from actual convolution. For example, attention weights are input-dependent, while convolution kernel parameters are independent of the input. In self-attention, the weights depend on the dot products between queries and keys, so different inputs will produce different attention distributions. The proposed method merely multiplies the attention weights by a fixed-shape mask; while the mask enforces locality, the attention values are still content-dependent, and thus, its translational equivariance may be less stable than that of convolution.
- Convolution not only enforces locality, but also weight sharing and a fixed linear combination pattern, both of which are important for feature stability. The proposed Learnable Mask employs a rolling mechanism to maintain consistency in the mask pattern. However, it still multiplies the mask pattern with attention weights that may vary significantly across positions, thereby ignoring the inherent variability of the weights themselves.
- One of ViT’s advantages over CNNs is the ability to model global dependencies. Applying a strong locality mask may suppress interactions between distant tokens, which could negatively affect tasks requiring full scene semantics.
- The experiments mainly compare against DeiT and ConViT, lacking a comprehensive comparison with other locality-enhanced Transformers (e.g., Swin Transformer).
- All experiments are conducted on classification tasks; there is no evaluation on detection or segmentation downstream tasks, leaving it unclear whether the proposed inductive bias would also be effective for tasks requiring precise spatial localization.

**Questions:**

The presentation of this paper should be largely improved.

---

### Official Review · Reviewer_nqUW · 2025-11-02

**Soundness:** 1
**Presentation:** 2
**Contribution:** 1
**Rating:** 0
**Confidence:** 4

**Summary:**

To improve the sample efficiency of vision transformers and enable them to perform effectively on small datasets, the authors propose adding a learned masks to the transformer to incorporate CNN-like inductive bias (spatial inductive bias). The method demonstrates positive results on ImageNet, outperforming the DeiT baseline while using fewer parameters and achieving faster performance.

**Strengths:**

- The paper is easy to follow.
- The results show improvement over the DeiT baseline on a well-known dataset such as ImageNet.
- The proposed method is simple to apply and can therefore be easily adopted by the community.

**Weaknesses:**

- The paper omits a significant amount of important related work. Several examples include Swin Transformer[1,2], Conv-based ViTs [3,4,5], and other works that incorporate inductive bias into vision transformers (for instance  MEGA [6], 2D-SSM [7], MaxViT [8], MixFFN and others). These works should be used both as baselines and to better differentiate the proposed method.
- There is no analysis of latency, FLOPs, or memory usage, during either training or inference.
- Further analyses are missing, such as evaluating robustness (Imagenet-A, Imagenet-E, and others) and beyond-classification performance (segmentation, generation).

___

**References:**

[1] Swin Transformer: Hierarchical Vision Transformer using Shifted Windows. ICCV 2021

[2] Swin Transformer V2: Scaling Up Capacity and Resolution. CVPR 2022

[3] CMT: Convolutional Neural Networks Meet Vision Transformer. CVPR 2022

[4] Early Convolutions Help Transformers See Better. NeurIPS 2021

[5] CoAtNet: Marrying Convolution and Attention for All Data Sizes. NeurIPS 2021

[6] Mega: Moving Average Equipped Gated Attention. ICLR 2023.

[7] 2D-ssm a general spatial layer for visual transformers. ICLR 2024

[8]  MaxViT: Multi-Axis Vision Transformer. ECCV 2022

**Questions:**

- Is there a reason for not comparing the proposed method with the methods in [1–8]?

- Where do the authors think this method can be applied, and what are the limitations of using distillation or other approaches for small datasets like [10]?

[10] Vision Transformer for Small-Size Datasets

---

### Meta-Review · Area_Chair_Q1eX · 2026-01-07

**Summary:**

All four reviewers recommend rejection, citing poor presentation, missing related work, weak novelty, insufficient experiments (limited to ImageNet-1K with outdated baselines, marginal gains, no true small-dataset evaluation), and theoretical shortcomings (proposed mask does not fully replicate CNN inductive biases).

**Reviewer Concerns:**

No author rebuttal was submitted, so none of the reviewers’ concerns were addressed. All concerns remain outstanding: missing comparisons to modern locality-biased ViTs (Swin, MaxViT, CoAtNet, etc.), lack of latency/FLOPs analysis, no downstream task evaluation, poor writing and organization, and incomplete theoretical justification.

**Reviewer Scores:**

Given the absence of rebuttal and the consistent critical issues raised, none of the reviewers are likely to change their scores:

- Reviewer nqUW: remains strong reject
- Reviewer 3zzV: remains reject
- Reviewer atqM: remains strong reject
- Reviewer V2QC: remains strong reject

---

### Decision · Program_Chairs · 2026-01-26

Reject